# Sustainable FDI in the Digital Economy

**Aneta Bobenič Hintošová * and Glória Bódy**

Faculty of Business Economy in Košice, University of Economics in Bratislava, 04130 Košice, Slovakia;
gloria.body@euba.sk
* Correspondence: aneta.bobenic.hintosova@euba.sk

**Abstract:** The shift towards a digital economy should lead to changes in the allocation methods of foreign direct investment (FDI), especially given the reduced need to transfer physical assets. At the same time, the need to understand and examine the sustainability of FDI as a relevant attribute throughout the life cycle of a given investment should be emphasized. The paper seeks to answer the research question whether more sustainable foreign direct investment is attracted in the digital economy. Hence, the paper explores the interlinks between the sustainability attributes of FDI and the development of the digital economy. For this purpose, a cluster analysis under the conditions of the countries of the European Union is conducted. The results of the cluster analysis carried out for the two periods show certain similarities, especially within the Nordic, Visegrad, Balkan, and Baltic groups of countries. The first group mentioned can be characterized by advanced digital development as one of the possible driving forces to attract sustainable FDI. The remaining groups show certain differences in this regard. Based on the results, the paper brings some policy implications towards emphasizing the sustainability attributes of foreign direct investment in the digital economy, especially through the implementation of the concept of sustainable investment promotion policy.

**Keywords:** sustainable foreign direct investment; digital economy; European Union countries; cluster analysis

## 1. Introduction

Success in attracting foreign direct investment (FDI) is generally considered one of the most meaningful indicators of a country's attractiveness and economic sustainability. However, FDI-led growth is also frequently associated with negative externalities, such as social inequalities, regional disparities due to the concentration of investors in better-developed regions, or environmental deterioration. The question of the quality of FDI, thus, seems to be more and more crucial. Hence, according to Sauvant and Gabor [1] governments should prioritize those FDI projects that are likely to foster the sustainable development of countries. Narula [2] suggested that to achieve long-term, sustainable growth it is important to encourage responsible and sustainable investment by adopting high environmental and social, as well as governance standards even in the initial phase of attracting FDI. This brings us to the concept of sustainable FDI, which can be seen as an investment responsible for the planet, people, and prosperity [3], thus, emphasizing the environmental, social, and economic effects of FDI. Although increased attention is drawn to the problem of FDI quality, it is still empirically underexamined in the literature.

Besides changing nature of foreign direct investment flows, the global economy is being fundamentally transformed by the onset of the digital economy gradually disrupting the existing economic order. This disruption has already started to modify the patterns of behavior of global multinational corporations, especially in the way of the reduced need of physical assets movement [4,5]. Hence, the development of the digital economy influences the patterns of foreign direct investment flows. According to Satyanand [3], it also supports sustainable development through more resource-efficient products, new green technologies, and technological inclusiveness accelerating global progress. However,

there is only a limited number of studies that address the issue of interlinks between foreign direct investment flows, especially from their sustainability point of view, and the development of the digital economy.

This paper is aspiring to contribute to the fulfillment of this gap in the existing literature by studying the interlinks between the sustainability attributes of FDI and the development of the digital economy. The paper follows this structure: the next part brings a review of the existing empirical literature focusing on the nexus between FDI and, more specifically, also sustainable FDI in the context of the digital economy followed by the description of the methodology applied within this study. The part related to findings is primarily based on the results of cluster analysis applied under the conditions of the 27 EU countries. The subsequent discussion of findings leads to conclusions including some policy implications for strengthening sustainable FDI in the digital economy.

## 2. Review of the Literature

To cover all aspects of the outlined topic, the literature review is focused first on the investigation of sustainability attributes of FDI, then on a review of studies examining the FDI–digitalization nexus, resulting in a look at sustainable FDI in the digital economy.

### 2.1. Sustainability Attributes of FDI

FDI generally occurs when a company makes a capital investment directly in facilities for the production or sale of goods or services in the host country, while this investment is usually accompanied by other material, technological, financial, information, or personnel flows [6]. The nature of accompanying flows is usually determined by the type of FDI performed. There are several classification aspects according to which the typology of FDI can be defined (for more details see, e.g., [7]), however, the reason for foreign engagement is usually underpinned by the strategic logic behind the following types of FDI [8]: resource-seeking FDI is driven by an access to labor supply, cheap raw material sources, infrastructure, etc.; market-seeking FDI is basically aimed at the penetration of local markets and related to factors such as market size and its potential growth, local market structure, etc.; efficiency-seeking FDI searches for new sources of competitiveness, economies of scale or specialization; in turn, strategic assets-seeking FDI enables the fostering of long-term strategic objectives of the investing company by, e.g., acquiring key assets of foreign companies, such as R&D capabilities, advanced technology, etc. As highlighted by Meyer [9], some FDI is made expressly to use assets acquired abroad to improve the investor's operations in other countries, particularly the investor's home country.

In addition, there is also another specifically motivated investment described in the literature, such as knowledge-seeking FDI, driven by an effort to improve technological or commercial capabilities based on learning. This is typical for the countries of the European Union as recipients of FDI from firms with origins in emerging economies, whose location choices are positively influenced by knowledge externalities [10]. There are also safety-seeking FDI linked mainly to risk-intolerant investors, who often come from countries at risk of expropriation and seek to invest in safe, mostly developed countries [11].

In light of the above FDI typology, more recent literature has come with the challenge to focus on investment, which contributes not only to the economic but also environmental and social development of host countries under a fair governance mechanism [12]. The attributes of FDI sustainability have been further developed by Kapuria and Singh [13], who empirically analyzed four dimensions, namely, the economic, social, environmental, and governance dimensions of sustainable FDI. However, to the best of the authors' knowledge, the current literature does not work with the concept of "sustainability-seeking FDI", which could be derived from the latest trends in balancing the economic, social, environmental, and governance attributes of FDI.

## 2.2. FDI in the Digital Economy

One of the first studies that highlighted the role of foreign direct investment in the context of the development of the digital economy was conducted by Zekos [14], who designated as target FDI locations those offering a cheap and highly skilled labor force, as well as clusters of information companies. Another insight into the FDI–digital economy nexus was provided by Gönel and Aksoy [15], who found that FDI inflows to information and communication technologies boost economic growth only in the case of a sufficient level of host countries' financial resources and human capital, as well as technological infrastructure. Eden [16] pointed to the fact that the digital economy brings new ways of doing business, which, to investment, means a reduced need for the presence of specific physical assets in host countries and, thus, a retreat of foreign direct investment. Subsequently, Ciuriak [17] also highlighted the differences in the motives to pursue inward foreign direct investment between traditional industrial or service sectors and a knowledge-based, data-driven economy. Karanina et al. [18], with regard to investing in the digital economy, added that the essence of direct investing is different due to the intangible nature of the property and its connection to the Internet.

However, the relationship between FDI and the development of the digital economy can also be explained from a bi-directional point of view. There are some studies (e.g., [19]) that attribute an active role to inward FDI in boosting the digital transformation of the economy, especially in the case of emerging and transforming economies. Similarly, Alibekova et al. [20] consider FDI flows to be an enabler of digital as well as innovation advancement of the economy of Kazakhstan. Banalieva [21] attributes an active role in this direction, especially to digital multinational corporations, which can contribute to the digital development of host countries.

However, in this context, Casella and Formenti [4] concluded that little systematic analysis has been conducted to examine investment patterns in the digital economy. According to Götz [22], even less is known about the digitally induced modification of investment promotion policies, especially with respect to desired engagement of new forms of FDI.

## 2.3. Sustainable FDI and the Digital Economy

Even less attention is paid to the interlinks between sustainable FDI and the digital economy development. One of the few studies in this regard studying these relationships partially is a study by Li et al. [23], who found an inverted U-shaped relationship between the development of the digital economy and $CO_2$ emissions, which is particularly true for developed economies. The early stages of digitization are accompanied by an increase in $CO_2$ emissions, but after reaching a certain peak of digital development, $CO_2$ emissions began to decrease. Moreover, the authors also highlighted the fact that different types of FDI influence $CO_2$ emissions in different ways. Subsequently, Khan et al. [24] investigated the interlinks between information and communication technology, ecological footprint, and economic complexity along with some other control variables including FDI and economic growth. The study conducted under the conditions of G7 countries (Canada, France, Germany, Italy, Japan, the UK, and the USA), among others, shows that a country's economic growth plays a detrimental role by reducing environmental sustainability, while foreign direct investment does not contribute to influencing the ecological footprint in the long term.

Luo et al. [25], in the Chinese context, showed that the development of the digital economy can indirectly foster the level of green innovation, for example by optimizing the industrial structure, increasing the degree of economic openness, or expanding the market potential. Moreover, the development of green innovations has a spatial spillover effect, since boosting green innovation in more developed regions may hinder green innovation in less developed regions due to the industrial transfers and talent flows.

Overall, Satyanand [3] pointed out the fact that countries can strategically use FDI to build and expand their digital economies on a basis of a strategic national digital development plan. This approach should actively also support sustainable FDI. The author

highlights especially the active role of investment promotion agencies in this regard, which should focus on mapping local talents, organizing targeted face-to-face or virtual meetings, creating innovation clusters and technology hubs, and helping to profile individual cities.

## 3. Materials and Methods

The main research objective of this study is to assess the interlinks among the EU countries in terms of sustainability attributes of their inward FDI, namely, economic, social, environmental, and governance attributes, and the development of their digital economy. This research objective led us to formulate the following research questions:

1.　Is more sustainable foreign direct investment attracted in the digital economy? Are there any differences among the EU countries in this regard?
2.　Is it possible to specify groups of the EU countries that show similarities in terms of sustainability attributes of FDI in the digital economy over more periods?

Since the research is exploratory in nature, no research hypotheses were formulated. When investigating sustainability attributes of FDI within the digital economy, we turned our attention to the 27 countries of the European Union. To capture different aspects of this possible interplay, we considered several indicators, which are presented in Table 1.

The selection of the presented indicators was influenced by the knowledge resulting from the literature review, as well as by the availability of input data. Since the object of our analysis were European Union countries of various sizes, we worked with relative indicators as well as with composite indexes that were suitable for comparability. Foreign direct investment is measured using the FDI net inflows as a percentage of GDP (similarly as in the majority of studies related to FDI such as [26]) as well as using inward FDI stock (that is also used in some studies, e.g., [27]). Since our main interest was to capture the sustainability attributes of FDI, in line with prevailing dimensions of sustainability (e.g., [13]), we used the following indicators: GDP per capita, GDP growth rate, and inflation rate to cover economic dimension, unemployment rate to capture economic and partially also social dimension, the Index of Economic Freedom to cover partially social and governance dimension, gender employment gap to capture the social and specifically gender aspect, and the Environmental Performance Index to cover an environmental dimension of sustainability. Moreover, we also used the World Happiness Index, as one the most popular composite indices of happiness, the sub-indicators of which can be linked to various aspects of sustainability [28]. The level of the development of the digital economy was considered through the Digital Economy and Society Index, which is specifically developed and applied, especially under the conditions of the European Union countries (e.g., [29]). When evaluating attributes that are relatively well captured by composite indicators (e.g., the Environmental Performance Index), we did not use further (individual) indicators to avoid their correlation.

For the purpose of data processing, we used hierarchical cluster analysis. It first considers each country as a small cluster and, based on the calculation of the distance between the clusters, merges the two closest clusters into a new larger one, so that the number of clusters decreases with each subsequent step. The measurement of the distance between the objects was based on Euclidean distance calculation. Ward's linkage was used to minimize the intra-group distances and maximize the inter-group distances in the clustering process. As a result, homogeneous clusters of countries were identified according to their similarities in terms of the indicators used.

The clustering of the countries was carried out in two periods: the first period included data on FDI for 2018 and other indicators for 2019 in order to capture the lagged effect of FDI on the rest of the indicators. Similarly, the second period covered data on FDI for 2021 and the rest of the indicators for the following year, i.e., the latest data at the moment of research completion. At the same time, such an approach allows us to reveal the possible effect of unexpected circumstances, such as the COVID-19 pandemic, on the aggregation of the EU countries within particular clusters. Table 2 shows the descriptive characteristics of the variables used for the two analyzed periods.

**Table 1.** Description of used indicators and their sources.

| Indicator (Symbol) | Description | Source |
|---|---|---|
| FDI inflow (FDIinflow) | Foreign direct investment net inflows as a percentage of gross domestic product. | World Bank [30] |
| Inward FDI stock (iFDIstock) | Inward foreign direct investment stock as a percentage of gross domestic product. | Eurostat [31] |
| GDP per capita (GDPpc) | Ratio of volume of a real gross domestic product to the average population of a particular country. | Eurostat [32] |
| GDP growth rate (GDPgr) | Annual growth rate of gross domestic product volume. | Eurostat [33] |
| Inflation rate (IR) | Inflation rate as annual average rate of change by harmonized index of consumer prices. | Eurostat [34] |
| Unemployment rate (UR) | Total unemployment rate from 15 to 74 years as a percentage of population in the labor force. | Eurostat [35] |
| Index of Economic Freedom (IEF) | Level of country's economic freedom measured by 12 principles of freedom—from property rights to financial freedom. | Heritage Foundation [36] |
| Environmental Performance Index (EPI) | State of country's sustainability measured in terms of climate change performance, environmental health, and ecosystem vitality. | Yale Center for Environmental Law and Policy [37], Socioeconomic Data and Applications Center [38] |
| Gender Employment Gap (GEG) | Difference between the employment rates of men and women aged 20–64. | Eurostat [39] |
| World Happiness Index (WHI) | Subjective well-being based on three main indicators: life evaluations, positive and negative effects. | United Nation's Sustainable Development Solutions Network [40] |
| Digital Economy and Society Index (DESI) | Digital progress of the EU countries measured by key digital areas: human capital, connectivity, integration of digital technology, digital public services. | European Commission [41,42] |

**Table 2.** Descriptive characteristics of used variables.

| Indicator | First Period | | | | | Second Period | | | | |
|---|---|---|---|---|---|---|---|---|---|---|
| | N | Min | Max | Mean | S.D. | N | Min | Max | Mean | S.D. |
| FDIinflow | 27 | −40.10 | 29.20 | −1.37 | 14.68 | 27 | −117.4 | 25.20 | 0.596 | 24.86 |
| iFDIstock | 27 | 17.00 | 5460.80 | 389.23 | 1092.91 | 27 | 20.50 | 4063 | 334.42 | 836.35 |
| GDPpc | 27 | 6630 | 83,590 | 27,714.81 | 17,447.59 | 27 | 7250 | 83,940 | 29,177.04 | 18,900.54 |
| GDPgr | 27 | 0.50 | 7.00 | 3.01 | 1.57 | 27 | −1.30 | 12.00 | 4.09 | 2.47 |
| IR | 27 | 0.30 | 3.90 | 1.68 | 0.95 | 27 | 5.90 | 19.40 | 10.73 | 3.70 |
| UR | 27 | 2.00 | 17.90 | 6.20 | 3.34 | 27 | 2.20 | 12.90 | 5.76 | 2.55 |
| IEF | 27 | 57.70 | 80.50 | 69.68 | 5.66 | 27 | 61.50 | 82.00 | 72.54 | 5.36 |
| EPI | 27 | 57.00 | 82.50 | 70.67 | 7.12 | 27 | 50.40 | 77.90 | 61.57 | 7.68 |
| GEG | 27 | 1.60 | 20.70 | 10.08 | 5.28 | 27 | 0.80 | 21.00 | 9.06 | 5.19 |
| WHI | 27 | 5.10 | 7.81 | 6.53 | 0.69 | 27 | 5.47 | 7.80 | 6.63 | 0.56 |
| DESI | 27 | 33.80 | 68.10 | 50.13 | 9.49 | 27 | 30.60 | 69.60 | 52.53 | 9.86 |

The descriptive characteristics of FDI show that while their average inflow in relation to GDP is, on average, negative in the first period, it slightly increases to positive figures

in the second period with more significant variation among the countries. The negative inflow of FDI is subsequently accompanied by a decrease in the average level of inward FDI stock between the two monitored periods.

In terms of GDP, both indicators increase within the two observed periods. However, notable variations in economic performance persist, as indicated by an even higher standard deviation in both cases, emphasizing the persistent presence of diverse economic conditions and outcomes among the countries in the second period.

As for the rate of inflation, there is a significant increase in the average inflation rate, together with a substantially higher degree of variation among the countries between the two periods. On the other hand, the average unemployment rate evolves in the opposite direction. It decreases slightly between the observed periods and similarly also diminishes the level of its variation among the monitored countries.

The average value of the Index of Economic Freedom increases between the two periods and the standard deviation decreases, which can indicate convergence of the countries of the European Union in the area of economic freedom. This also applies to the social aspect of sustainability, as there are clear shifts in the range and mean values of the gender employment gap, indicating that there is a slight decline in the gender inequality in the labor market in terms of employment across the EU.

The observed variations in the Environmental Performance Index between the two periods show rather negative changes in environmental performance, accompanied by a decrease in the mean EPI values and a slight increase in variation among the EU countries. On the other hand, the mean values, as well as the standard deviation of the World Happiness Index develop exactly in the opposite direction between the two observed periods. The level of happiness, thus, increases on average across the EU, despite a slight deterioration in environmental performance and a deepening of differences among the EU countries in this regard.

The observed changes in the Digital Economy and Society Index over these two periods indicate slightly positive shifts in the level of digital development, as well as an increasing level of variation of the EU countries in terms of their digitization.

Before conducting the cluster analysis, correlations using the Pearson coefficient between each pair of variables for both periods were separately checked. The correlation matrix is shown in Table 3. The values above the diagonal show correlation coefficients in the first period, while the values under the diagonal show correlation coefficients in the second period.

**Table 3.** Correlation matrix.

| | FDIinflow | iFDIstock | GEG | DESI | EPI | IEF | WHI | UR | IR | GDPgr | GDPpc |
|---|---|---|---|---|---|---|---|---|---|---|---|
| FDIinflow | 1 | −0.408 * | 0.216 | −0.060 | −0.155 | −0.096 | −0.157 | 0.097 | −0.304 | 0.240 | −0.304 |
| iFDIstock | −0.351 | 1 | 0.073 | 0.090 | 0.271 | 0.221 | 0.205 | −0.074 | −0.079 | 0.135 | 0.614 ** |
| GEG | −0.118 | −0.010 | 1 | −0.482 * | −0.153 | −0.396 * | −0.211 | 0.235 | −0.048 | 0.245 | −0.149 |
| DESI | 0.084 | 0.174 | −0.504 ** | 1 | 0.670 ** | 0.705 ** | 0.809 ** | −0.188 | −0.186 | −0.234 | 0.610 ** |
| EPI | 0.108 | 0.318 | −0.333 | 0.715 ** | 1 | 0.352 | 0.857 ** | 0.066 | −0.370 | −0.565 ** | 0.805 ** |
| IEF | −0.018 | 0.312 | −0.584 ** | 0.690 ** | 0.480 * | 1 | 0.650 ** | −0.525 ** | 0.183 | 0.049 | 0.549 ** |
| WHI | 0.085 | 0.123 | −0.361 | 0.739 ** | 0.715 ** | 0.637 ** | 1 | −0.274 | −0.140 | −0.352 | 0.755 ** |
| UR | −0.100 | −0.155 | 0.191 | −0.045 | −0.146 | −0.421 * | −0.185 | 1 | −0.498 ** | −0.343 | −0.048 |
| IR | 0.181 | −0.236 | −0.214 | −0.345 | −0.405 * | 0.104 | −0.261 | −0.192 | 1 | 0.247 | −0.345 |
| GDPgr | −0.043 | −0.051 | 0.423 * | −0.007 | −0.211 | −0.167 | −0.216 | 0.044 | −0.426 * | 1 | −0.293 |
| GDPpc | −0.050 | 0.541 ** | −0.160 | 0.649 ** | 0.509 ** | 0.611 ** | 0.667 ** | −0.117 | −0.499 ** | 0.180 | 1 |

Correlations in the first period are above the diagonal, in the second period are under the diagonal. *, ** correlation is significant at the level of 0.005 or 0.001, respectively (two-tailed).

In line with prevailing recommendations also applied in other studies (e.g., [43]), we removed those variables that demonstrated strong (i.e., strength of the relationship above 0.75) statistically significant (at $p < 0.05$) relationships with other variables. Hence, the World Happiness Index and GDP per capita were not included in the cluster analysis for the first period. To achieve comparable results, the cluster analysis for the second period was performed both without these two variables and additionally with the full data set. To meet the requirements of Euclidean distance calculation, all the variables were standardized.

## 4. Results

The results of the clustering of 27 EU countries for the first period, based on nine indicators (i.e., excluding the World Happiness Index and GDP per capita) are reported in the dendrogram (Figure 1) using Ward´s linkage.

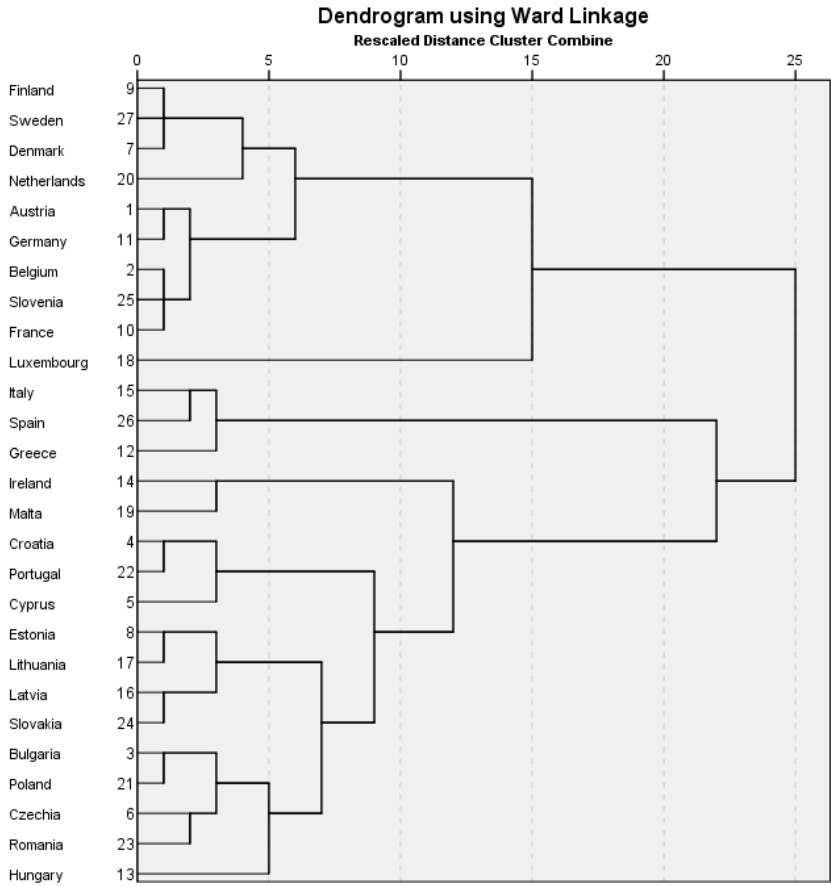

**Figure 1.** Hierarchical cluster analysis for the first period—without WHI and GDPpc.

The cluster analysis reveals the existence of four clusters. For their better understanding, we also present the means of the used indicators for each identified cluster separately (Table 4).

Countries belonging to the first cluster receive a small positive portion of FDI related to their GDP in the monitored year, which amount is reflected in the relatively low stock of inward FDI. On the other hand, countries in this cluster are European digitization leaders. Similar results can be concluded with regard to the social aspect of sustainability (the lowest gender employment gap, high level of economic freedom, relatively low unemployment rate), but also the environmental aspect with the second highest value of the Environmental Performance Index.

The second cluster is formed only by Luxembourg, which reports extreme values in many aspects, namely, negative FDI inflow and the highest overall inward FDI stock. It is

accompanied by the best values of several indexes related to environmental performance and economic freedom.

**Table 4.** Mean values of indicators in four specified clusters—first period.

| Indicator | Cluster 1—Mean<br>Finland, Sweden, Denmark, Netherlands, Austria, Germany, Belgium, Slovenia, France | Cluster 2—Mean<br>Luxembourg | Cluster 3—Mean<br>Italy, Spain, Greece | Cluster 4—Mean<br>Ireland, Malta, Croatia, Portugal, Cyprus, Estonia, Lithuania, Latvia, Slovakia, Bulgaria, Poland, Czechia, Romania, Hungary |
|---|---|---|---|---|
| FDIinflow | 0.04 | −36.70 | 2.70 | −0.2 |
| iFDIstock | 99.23 | 5460.80 | 30.87 | 290.17 |
| GEG | 6.86 | 9.10 | 17.33 | 10.66 |
| DESI | 57.67 | 54.50 | 43.43 | 46.20 |
| EPI | 77.50 | 82.30 | 71.47 | 65.27 |
| IEF | 71.74 | 75.90 | 61.87 | 69.59 |
| UR | 5.48 | 5.60 | 13.97 | 5.04 |
| IR | 1.48 | 1.60 | 0.63 | 2.04 |
| GDPgr | 1.88 | 2.30 | 1.47 | 4.12 |

Italy, Spain, and Greece belong to the third cluster, for which low inward FDI stock but relatively higher positive FDI inflow is typical. The less intensive presence of foreign investors in these countries seems to be related to a weaker social dimension, which is reflected in the highest unemployment rate, obvious gender differences in employment, and lower level of economic freedom. These countries show the lowest growth rate of GDP, but also the lowest inflation rate. Their level of digitization is considerably below the EU average.

The last cluster consists especially of newly acceding EU countries, i.e., the Visegrad, Baltic, and Balkan countries supplemented by Ireland, Malta, Portugal, and Cyprus. These countries have a relatively high stock of inward FDI, although accompanied by a relatively low inflow in the monitored year. These countries show the highest GDP growth, which may be a sign of an effort to catch up economically with the "old" EU countries, also with the assistance of incoming FDI. This investment activity could be manifested in the low unemployment rate and sufficient results in other socially oriented indicators. On the other hand, these countries perform worse in the Environmental Performance Index. The level of their digitization is below the EU average, but better than in the countries from the previous cluster.

Further, it was in our interest to look at the composition of countries within particular clusters recently, considering the latest available data. Similarly, the clustering of 27 EU countries for the second period, first excluding the World Happiness Index and GDP per capita, was performed. The results are reported in the dendrogram (Figure 2) using Ward's linkage.

In Figure 2, four clusters can be identified again, but their composition is slightly different than in the previous case. Namely, Ireland and Malta move from cluster four to cluster one and, thus, are more linked with the majority of countries of the "old" EU. However, Belgium and France move to cluster four. The Baltic countries leave cluster four and form a separate cluster two. Cyprus joins Luxembourg and they form cluster three. In the second period, Greece, Italy, and Spain join the remaining countries that are originally in cluster four.

Interestingly, the three EU countries belonging to the G7 group do not appear in the same cluster and, thus, do not show obvious similarities in any of the two periods.

Moreover, the only country from the G7 that remains clustered with European digitization leaders is Germany.

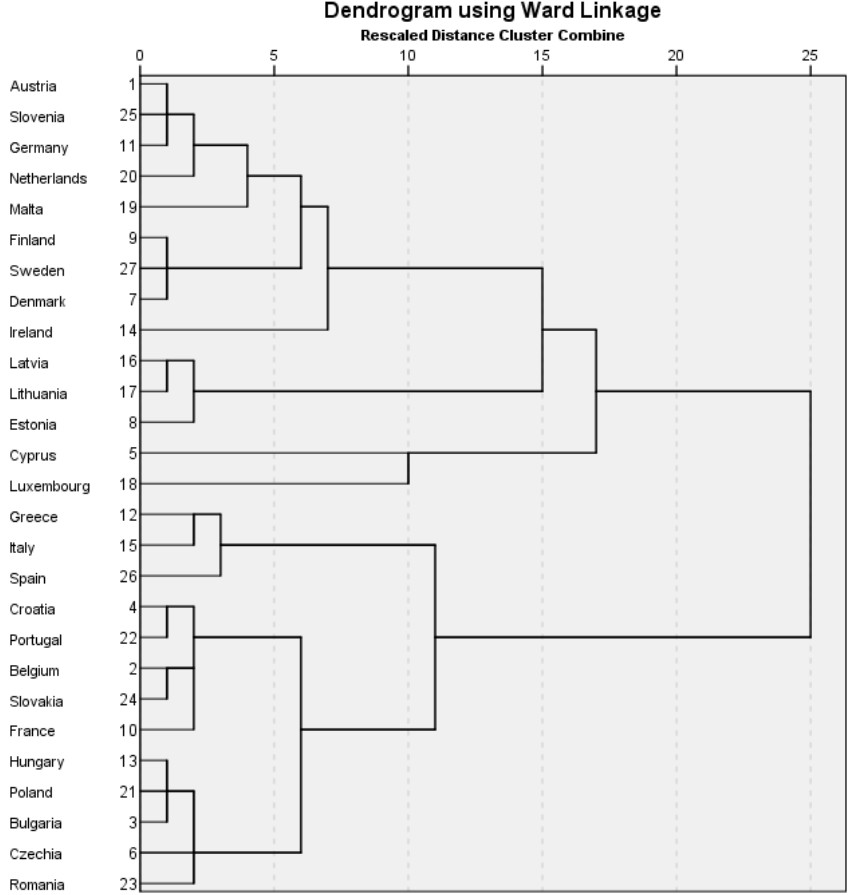

**Figure 2.** Hierarchical cluster analysis for the second period—without WHI and GDPpc.

Since the values of all indicators in the second period met the correlation restrictions, we also performed the cluster analysis with the full data set. Results are reported in Figure 3. Again, four clusters with the same composition as in the previous case can be identified. Hence, the World Happiness Index together with GDP per capita do not cause changes in the structure of particular clusters.

For a better understanding of the composition of each cluster, we also present the means of the indicators used for each identified cluster separately (Table 5). Countries marked in bold have changed their position in individual clusters compared to the first period.

The first cluster is formed mainly by the Visegrad and Balkan countries, which are joined also by Belgium, France, Greece, Italy, and Spain in the second period. This group of countries shows the lowest portion of inward FDI stock accompanied by the highest unemployment rate and a gap in employment with regard to gender. The lowest values of economic freedom, happiness, environmental performance, and digitization are also obvious.

The Baltic countries are grouped into a separate cluster two, with the highest positive values of current inflows of FDI, together with the highest level of economic freedom, but also inflation rate. This group of countries reports the lowest gender employment rate, but also GDP growth, as well as GDP per capita.

The third cluster consists mainly of the "old" EU countries, with relatively higher inward FDI stock and the highest level of economic growth, environmental performance, and happiness, as well as digitization.

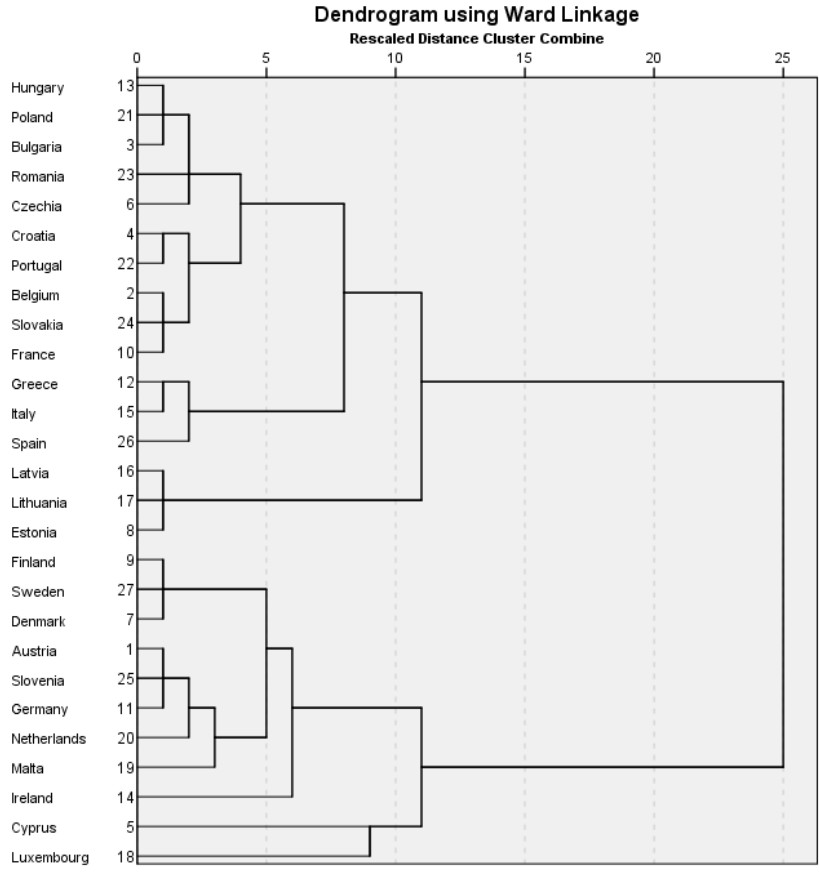

**Figure 3.** Hierarchical cluster analysis for the second period—full data set.

**Table 5.** Mean values of indicators in four specified clusters—second period.

| Indicator | Cluster 1—Mean | Cluster 2—Mean | Cluster 3—Mean | Cluster 4—Mean |
|---|---|---|---|---|
| | Hungary, Poland, Bulgaria, Romania, Czechia, Croatia, Portugal, **Belgium,** Slovakia, **France, Greece, Italy, Spain** | **Latvia, Lithuania, Estonia** | Finland, Sweden, Denmark, Austria, Slovenia, Germany Netherlands, **Malta, Ireland** | **Cyprus,** Luxembourg |
| FDIinflow | 4.27 | 11.17 | 6.12 | −64.00 |
| iFDIstock | 65.49 | 70.27 | 260.19 | 2812.65 |
| GEG | 11.70 | 2.27 | 7.47 | 9.30 |
| DESI | 45.85 | 52.97 | 61.79 | 53.65 |
| EPI | 56.56 | 59.47 | 68.72 | 65.15 |
| IEF | 68.22 | 76.87 | 76.40 | 76.75 |
| WHI | 6.31 | 6.48 | 7.12 | 6.68 |
| UR | 6.47 | 6.17 | 4.62 | 5.70 |
| IR | 10.90 | 18.50 | 8.47 | 8.15 |
| GDPgr | 4.29 | 1.13 | 4.90 | 3.50 |
| GDPpc | 19,714.62 | 14,846.67 | 41,828.89 | 55,245.00 |

Cyprus and Luxembourg forming the fourth cluster reach extreme values in some aspects. Despite their negative FDI inflow in the monitored year, they have the highest stock of inward FDI, which can be associated with the highest GDP per capita, as also confirmed by the overall positive statistically significant correlation between the inward

FDI stock and GDP per capita in both periods. The values of the rest of the indicators oscillate around the EU average or slightly above it.

## 5. Discussion

Although the European Union is generally considered as a group of developed countries that should be at advanced stages of the investment development path based on Dunning´s theory [44], there are some variations between the EU countries. Sonderman and Vansteenkiste [45] showed that the introduction of the single currency changed the driving forces of FDI inflows in the EU, namely, the Euro began to facilitate vertical FDI flows within the Eurozone, but also to support horizontal FDI flows from outside the Eurozone. In addition to these changes in FDI flows within the EU, there have also been gradual shifts in extra EU FDI flows. Witkowska [46] pointed to unfavorable changes in the investment climate, resulting in serious divestments between the US and the EU. As a result, Canada and Switzerland became the main bearers of FDI into the EU. Moreover, another trend is also apparent, namely, the increase in investment in the EU from emerging economies, as pointed out by Jindra et al. [10]. Thus, changes in the distribution of inward FDI within the EU, as well as in their effects from a sustainability point of view, can be attributed to the above-mentioned trends. Similar conclusions can be stated with regard to the level of the digital development of economy and society of the EU countries. In its DESI report, the European Commission [42] also notes that there are still large gaps between the EU leaders and the countries with the lowest DESI scores, although the level of divergence between member states is diminishing.

Even when looking at the three EU countries that belong to the G7 group, namely, France, Germany, and Italy, it is clear that they do not form a homogeneous group, as they do not appear in the same cluster in any of the monitored periods. Moreover, Khan et al. [24] show in their study that despite advances in knowledge and digitization, the G7 countries are responsible for environmental unsustainability, the reduction in which does not seem to be supported by foreign direct investment or business activities. However, our results do not prove such conclusions, and we believe that groups of countries such as the EU or G7, grouped mainly based on political or economic criteria, should not be considered homogeneous when evaluating other aspects such as digitization or ecology. Moreover, there are increasing waves of criticism towards traditional informal organizations such as the G7 pointing to the increasing contestation of the world order and the rising power of countries such as China or India [47].

Hence, it is reasonable to look at the EU countries through a narrower lens and find relatively homogenous groups in terms of key aspects of our research, namely, sustainable FDI and the development of the digital economy. Based on the conducted cluster analysis, it seems that groups of countries receiving a relatively higher portion of FDI and, thus, having foreign investors operating in their countries (when looking especially at inward FDI stock) show, on average, lower level of unemployment rates and relatively higher growth of GDP, but also inflation rates. Hence, FDI can generally have important economic effects as it has already been reflected especially within the FDI-led growth hypothesis, the validity of which was confirmed by several studies conducted across Europe (e.g., [48–50]).

Similarly, countries with higher levels of inward FDI stock have a relatively lower gender employment gap, higher economic freedom, and are happier, which can indicate positive effects on the social dimension of sustainability and vice versa. The second may be the case of Greece, Italy, and Spain, and to some extent also the Visegrad and Baltic countries, where foreign direct investment has focused mainly on industrial sectors with job opportunities preferably for men. This widens the gap in the employment between women and men, which probably contributes neither to the level of happiness nor to economic freedom in these countries. These considerations can partially be underpinned by the results of the study by Vahter and Masso [51] or also the study by Magda and Sałah [52], who examined differences in gender wage gaps between foreign-owned firms in Poland.

The interaction between inward FDI and environmental dimension seems to support the pollution halo effect hypothesis, which was in the case of developed countries confirmed, e.g., by the study of Abid and Sekrafi [53]. We found that there are groups of countries (especially the Nordic countries) performing environmentally better and receiving a relatively higher portion of FDI on one hand, as well as groups of countries (especially the Balkan, Baltic, and Visegrad countries) whose inward FDI stock decreased between the two monitored periods and their environmental performance worsened as well. Hence, countries that aspire to attract more foreign investment and deepen positive environmental spillover effects of FDI should revise their investment promotion policies in order to be more sustainable. Some recommendations to attract cleaner and high-quality foreign investment based on the identified pollution halo effect of FDI can be found in the study by Gao et al. [54]. In order to select the most appropriate instruments to support sustainable investment, more targeted research considering parameters such as climate change and greenhouse gas emissions would be desirable.

When looking in more detail at the investment promotion policy applied in the Visegrad countries, it is clear that the social effects of investment are strongly represented there. This is particularly supported by the provision of grants for the creation of new jobs and/or trainings [55], although often with questionable efficiency (for more details see [56]). On the other hand, it is difficult to find any specific environmentally oriented grants that could be motivation for attracting green foreign investment. The existing scheme of investment incentives can also be supplemented with environmentally oriented indicators, which a specific investor would have to fulfill when applying for the provision of investment incentives. More sustainable FDI could, thus, be attracted and a concept of sustainable investment promotion policy established.

There are also some interlinks between inward FDI and digitization that deserve further investigation. The results of conducted cluster analysis indicate that groups of countries with a relatively higher stock of inward FDI and, thus, with a more intensive presence of foreign investors, report higher scores of the Digital Economy and Society Index. However, particularly in the Nordic countries, it can be explained by the possible investments related to advanced companies and technologies [57] and, thus, digital development can be considered as the driving force of knowledge and technology-seeking investment. On the other hand, the Visegrad, Baltic, and Balkan countries can instead use inward investment as a means for their further digital development. Hence, a deeper study of these potential FDI–digital economy bi-directional relationships can form the agenda of future research. In this regard, the structure of inward FDI, especially in terms of their sectoral composition, should be considered. For this purpose, those sectors that were not only attractive to foreign investors, but also significantly affected by digitization, can be considered. A good example in this regard is a sector of payment services that have witnessed significant digitization efforts, especially through the adoption of FinTech solutions worldwide (for more details see, e.g., [58,59]).

Looking at the interplay between sustainable FDI and digital development, the Nordic countries clustered with some other "old" EU countries and Slovenia could form a good example of linking digital development and inward investment activity with positive effects on the economy, society, and environment, resulting in the highest level of national happiness. Stephenson´s study [60] offers some other interesting recommendations to boost investment activity in parallel with the digital transformation, leading to the achievement of the goals of sustainable development.

## 6. Conclusions

The present study aimed to assess the interlinks among the EU countries in terms of the range of sustainability attributes of FDI and the development of their digital economy.

Our review of the literature with regard to the types of FDI according to the investors' motives shows that a comprehensive concept of sustainability-seeking FDI has not yet been elaborated in the current literature, which represents an interesting theoretical and

empirical future research challenge. As part of our analysis, we subsequently worked with the flow and stock of inward FDI and their sustainability attributes, which found support in the current empirical literature, namely, economic, social, environmental, and governance dimension. Each of the dimensions was captured by a set of relevant indicators, with certain overlaps between individual dimensions. Moreover, we also considered the level of the digital development using the Digital Economy and Society Index. To achieve the research objective, cluster analysis using Ward's linkage was applied under the conditions of the EU countries.

In searching for answers to the research questions, we found that there are some countries grouped in a particular cluster (especially around the group of Nordic countries) in which their advanced digital development can be considered as one of the driving forces for attracting more sustainable foreign direct investment. However, there are differences among particular clusters in this regard. Hence, more light could be shed on this issue in the future more detailed analysis.

For this purpose, the conducted cluster analysis reveals relatively homogenous groups of the EU countries that may be subject to more detailed observations. In this regard, as homogenous those groups of countries can be considered that remain together in the particular clusters in both periods and, thus, report substantial similarities, namely: group one—the Visegrad countries (Czechia, Hungary, Poland, Slovakia), together with the Balkan countries (Croatia, Bulgaria, Romania) and Portugal; group two—Greece, Italy, and Spain; group three—the Baltic countries (Latvia, Lithuania, Estonia); group four—Finland, Sweden, Denmark, Austria, Slovenia, Germany, and the Netherlands. Deeper future research under conditions of these countries, using, e.g., panel data analysis in order to reveal further relationships together with a contextual analysis of current investment promotion policy settings, could bring interesting insights to pursue appropriate investment promotion policies toward more sustainable FDI.

In summary, it was not our primary intention to provide ready-made answers to research questions that may arise in relation to sustainable investments in the digital economy, but rather to stimulate further discussion in this relatively under-researched area and to outline possible future research directions.

**Author Contributions:** Conceptualization, A.B.H.; methodology, A.B.H.; formal analysis, A.B.H. and G.B.; resources, A.B.H. and G.B.; data curation, G.B.; writing—original draft preparation, A.B.H.; writing—review and editing, A.B.H. and G.B.; visualization, G.B. All authors have read and agreed to the published version of the manuscript.

**Funding:** This research received no external funding.

**Institutional Review Board Statement:** Not applicable.

**Informed Consent Statement:** Not applicable.

**Data Availability Statement:** The data used are presented in References section under items [30–42].

**Conflicts of Interest:** The authors declare no conflict of interest.

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
