# Peer review of "Sustainable FDI in the Digital Economy"

_sustainability, doi:10.3390/su151410794_

Round 1

Reviewer 1 Report

This paper describes the sustainable foreign direct investment (FDI) in the digital economy by using cluster analysis within the Nordic, Visegrad, Balkan and Baltic groups of EU countries. 

Looking at the correlation matrix using the pearson coefficient shows that in some countries, there exists a positive correlation between FDI and digitization.   

The research title is very interesting and some further insight can be gained by answering these questions:

1) What parameters are used to describe the level of digital economy achieved by each country?

2) Is it possible to draw radar plots to show example countries from each cluster with the highest ranking of attributes pointing towards the development of digital economy and .

3) Why all the EU countries with the highest correlation of digital economy and FDI are not comparable to G7 economies?

4) Can sustainable investments be guaranteed without looking into other parameters like climate change and increasing green house gases (GHG) by major digital economies?

5) What time horizon of sustainability is in focus for this research? How many years can be regarded as long term or short term.

Some typos need correction, For example:

1) Hence, (in) future more detailed analysis could shed more light on this issue (missing word in bracket)

2) 'This is typical for the countries of the European Union as recipients of FDI from firms from emerging economies' should be modified for clarity.

and so on.

Minor editing required.

Reviewer 2 Report

Interesting topic related to FDI and digitalization. However, i have observed some issues;

(1) The title is too short and does not reflect proper insight that may attract the readers. 

(2) What is the process of cluster analysis and how data is collected and used not clear from the paper.

(3) the research objective is not clear? 

(4) the paper is based on EU and how FDI and digitalization are linked where the EU is already a developed union. 

(5) what are the sources of FDI in EU? they are among EU members or outside and what is the share?

(6) What is the role of FInTECH and banks in Digitlization and FDI? this paper might be useful; Dwivedi, P., Alabdooli, J.I. and Dwivedi, R., 2021. Role of FinTech adoption for competitiveness and performance of the bank: A study of banking industry in UAE. International Journal of Global Business and Competitiveness16(2), pp.130-138.

(7) what is the development taking place with Digitlization and FDI in EU and what is the ground relaity? as india is doing great in FDI and Digitlization and a great case to reflect upon: Dwivedi, R., Alrasheedi, M., Dwivedi, P. and Starešinić, B., 2022. Leveraging financial inclusion through technology-enabled services innovation: A case of economic development in India. International Journal of E-Services and Mobile Applications (IJESMA)14(1), pp.1-13.

Please address the issue

okay

Reviewer 3 Report

The topic of the paper is relevant and very interesting. Authors highlighted three important topics: sustainability, FDI and digital economy. They used a wide range of international literature sources and cited them correctly. Most part of them are from the last few years. Their paper’s main question whether more sustainable foreign direct investment is attracted in the digital economy. In the paper they introduced the interlinks between the sustainability attributes of FDI and the development of the digital economy. They formulated 2 research questions and chose 11 indicators. In the cluster analysis authors separated the 27 EU countries well. They illustrated results by using clear figures and tables. As a conclusion we can say that this paper can give us few policy implications towards emphasizing the sustainability goals of FDI in our digital world.   I appreciate their findings and results.

Reviewer 4 Report

 This manuscript contributes to the field of studying the interlinks between the sustainability attributes of FDI and the development of the digital economy. The manuscript is well-organized, and the results are promising in attracting more research in this interesting field.

 I have the following comments.

- Line 7: Abstract:

The Abstract addresses only a few parts of the results. Therefore, I recommend extending the Abstract by adding an overview of the results and research implications.

- Line 151: “…sustainable FDI, i.e. FDI that is responsible for the planet, people, and prosperity.”

I recommend expanding and moving this clear definition of sustainable FDI to an earlier section of the manuscript. It is helpful to give a clear definition of sustainable FDI as early as possible (because this is a core concept/definition).

- Line 176: “Since our main interest was to capture the sustainability attributes of FDI, in line with prevailing dimensions of sustainability (e.g. [13]), we used the following indicators: GDP per capita, GDP growth rate, and inflation rate to cover economic dimension, unemployment rate to capture economic and partially also social dimension, Index of Economic Freedom to cover partially social and governance dimension, gender employment gap to capture the social and specifically gender aspect, and Environmental Performance Index to cover an environmental dimension of sustainability.”

 It is noticed that the four (4) dimensions of FDI sustainability (namely, the economic, social, environmental, and governance dimensions of sustainable FDI) are not evenly (or equally) represented by the indicators used in this study. For example, more indicators cover the economic dimension (4 indicators) compared to one (1) indicator only covering the governance dimension. Could you please comment on this observation and its effect on the overall results? What other indicators can be added to cover the governance dimension?

- Please review the English and grammar (the manuscript needs to be proofread as it contains some typos and grammatical errors).

 Moderate editing of the English language is required

Round 2

Reviewer 2 Report

Thanks for addressing the comments and now the paper is in better and logical.

okay